# Use of Collagen in Cosmetic Products

**Barbara Jadach** [1],*  , **Zofia Mielcarek** [2] and **Tomasz Osmałek** [2]

1   Division of Industrial Pharmacy, Chair and Department of Pharmaceutical Technology, Faculty of Pharmacy, Poznan University of Medical Sciences, 3 Rokietnicka, 60-806 Poznan, Poland

2   Chair and Department of Pharmaceutical Technology, Faculty of Pharmacy, Poznan University of Medical Sciences, 3 Rokietnicka, 60-806 Poznan, Poland; zosiam.66@wp.pl (Z.M.); tosmalek@ump.edu.pl (T.O.)

*   Correspondence: bajadach@ump.edu.pl; Tel.: +48-664159558

**Abstract:** Collagen (CLG) belongs to the family of fibrillar proteins and is composed of left-handed α polypeptide chains, which, twisting around themselves and their axis, form a right-handed superhelix. In the chemical structure, it contains mainly proline, hydroxyproline, glycine, and hydroxylysine. It occurs naturally in the dermis in the form of fibers that provide the skin with proper density and elasticity. The review aimed to present the types of collagen protein, factors affecting its structure and its unusual role in the functioning of the human body. Also, an overview of cosmetic products containing collagen or its derivatives, the characteristics of the formulas of these products, and the effects of their use were presented. Throughout the market, there are many cosmetic and cosmeceutical products containing CLG. They are in the form of fillers administered as injections, belonging to the group of the oldest tissue fillers; products administered orally and for topical use, such as creams, gels, serums, or cosmetic masks. Analyzed studies have shown that the use of products with collagen or its peptides improves the general condition of the skin and delays the aging process by reducing the depth of wrinkles, improving hydration (in the case of oral preparations), reducing transepithelial water loss (TEWL), as well as improving skin density and elasticity. In addition, oral application of bioactive CLG peptides has shown a positive effect on the nails, reducing the frequency of their breakage.

**Keywords:** collagen; skin; cosmetics; cosmeceutical products





## 1. Introduction

The cosmetics industry is one of the best-developing industries in recent years [1,2]. The cosmetics market offers countless products in various formulas so that everyone can find a cosmetic that suits their needs. Consumer awareness of the composition of cosmetic products is also regularly increasing, to which companies producing cosmetics and cosmeceuticals respond by offering a wide range of active ingredients that act on specific skin properties [1]. The main purpose of the cosmetic is to improve the appearance. The term cosmeceutical describes a cosmetic with a healing effect due to the supposedly higher content of active ingredients. In the eyes of the law, it is subject to the same legal regulations as a cosmetic product [2,3]. One of the most frequently used ingredients in cosmetics is collagen (CLG) [4,5]. It is a structural protein composed of amino acids that create collagen fibers, characterized by exceptional strength and high elasticity [6,7]. This protein is composed of three left-handed α polypeptides that wind around themselves and their axis to form a right-handed superhelix [8]. Its structure varies depending on its functions and place of occurrence. CLG is one of the most important proteins in the human body because it is responsible for maintaining the appropriate structure of tissues and organs and constitutes as much as one-third of the total body protein mass [7]. It occurs, among the main organs in the body that provide appropriate elasticity and strength [8–11]. It is also an essential building block of the skin; without CLG, it would not be able to perform its functions properly. CLG proteins in the dermis are constantly being rebuilt, and

their fibers are particularly resistant to stretching and tearing [12–14]. It owes its success in the cosmetics industry to skin biocompatibility, as it is a natural skin component [5,15]. With age, the production of natural collagen decreases, which is why producers of cosmetics and cosmeceuticals offer products that allow for replenishment of this ingredient both from the inside and outside [5,16]. The cosmetics industry is constantly developing and looking for new, innovative solutions, so sources of obtaining collagen for cosmetic products have become the subject of many studies. Currently, the best tested and used CLG in cosmetics is of marine origin [3,6,16–19]. Among the CLG products used for cosmetic and aesthetic purposes, one can distinguish collagen fillers, which belong to the group of the oldest tissue fillers; products with CLG or its hydrolysates for topical use on the skin; and oral products [3,19,20]. Each of these forms is intended to somehow replenish the loss of this protein in the skin. Taking into account the fact that CLG is a large-molecule protein and is unable to cross the epidermal barrier, its hydrolysates, i.e., smaller peptide fragments with a lower molecular weight, are used in topical products [5,21,22].

This work aims to show the unusual structure of collagen protein, its irreplaceable role in the skin, but also in the entire body, which makes it a very interesting ingredient of cosmetic products with various mechanisms of action. This paper presents the types of cosmetics containing CLG or its derivatives, in what form it occurs, and how it affects the properties of the skin, as well as the effects of its action confirmed by research.

## 2. Characteristics of Collagen

### 2.1. Structure and Biosynthesis

Collagens are family of fibrillar proteins that dominate the extracellular matrix of most connective tissues in mammals [23,24]. They are the basic proteins responsible for the structure and biochemical properties of connective tissue, and in the case of skin, they constitute approximately 70% of its dry weight. These are natural polymers constituting 1/3 of the total mass of proteins in the human body, mainly performing building functions [7]. They are characterized by a unique form that varies depending on their function and place of occurrence [23,25]. CLG commonly found in organisms is type I. Its presence, e.g., in bone tissue, provides elasticity and strength. Collagens are a fibrous components of the skin, tendons, ligaments, cartilage, and blood vessel walls. They are also present in bones and teeth and internal organs such as the heart, lungs, and liver [7,23].

In terms of structure, collagens are among the most complex natural polymers [8,12–14,26,27]. More than 20 amino acids may be involved in the construction of these protein molecules. Regardless of the type of CLG, the most abundant amino acids are proline (Pro), hydroxyproline (Pro-OH), glycine (Gly), and hydroxylysine (Lys-OH). The unique structure of CLG proteins consists of three left-handed polypeptide α chains, the so-called procollagen. They are wrapped around each other and have a common axis and thus form a right-handed conformation of a superhelix called tropocollagen [8,12,23].

The superhelix is held together by hydrogen bonds, electrostatic interactions, and van der Waals forces [8]. In addition to three-helical domains, non-helical fragments, i.e., telopeptides, are also involved in the structure of collagens (Figure 1a). They occur at the ends of CLG macromolecules or are built into superhelix structures. The presence of a triple helix and its content ranges from 96% in type I CLG to less than 10% in type XII CLG [8,23]. The fibers of most types of collagen form delicate networks, except types I and II, whose fibers are thick and resistant to stretching. Natural collagen superhelices are resistant to proteases such as pepsin, trypsin, and chymotrypsin [14,23,27,28].

The only enzymes that can degrade these macromolecules are called collagenases; they are extracellular matrix metalloproteinases [29]. Each chain contains about 1000 amino acids. A characteristic feature of the amino acid composition of CLG polypeptides is the equimolar amount of acidic amino acids, i.e., glutamic acid (Glu) and aspartic acid (Asp), and basic amino acids, i.e., lysine (Lys) and arginine (Arg) [29]. The most frequently repeated sequence in the collagen polypeptide chain (Figure 1b) is Gly-X-Y-, of which Gly constitutes the third amino acid residue, and X and Y are the amino acid residues:

Pro (approx. 28%) and Pro-OH (approx. 38%) [8,12,29]. The number of kinks occurring in the collagen structure, resulting from disruptions in the regularly repeating Gly-X-Y sequence, determines its elasticity. Better aggregation possibilities are provided by greater flexibility, e.g., in membrane structures. The unique structure of the triple helix and strong bonds between amino acids make collagen fibers maintain their flexibility and are resistant to stretching. Collagen proteins are encoded by many genes, which is why they are structurally and functionally heterogeneous [29]. Thanks to this, organisms protect themselves against the loss of these important macromolecules. Also, changes occurring during post-translational modification are important in maintaining the heterogeneity of collagen proteins, which are constantly synthesized and degraded in the extracellular space [12]. This means that the vast majority of CLG fibers are made of more than one type of CLG [12,14]. Before CLG appears in the extracellular space, it undergoes some modifications like hydroxylation, glycosylation, and the process of creating triplet procollagen particles. The resulting procollagen is expelled outside the cell. Then, using peptidases, telopeptides are removed and tropocollagen molecules are formed [12,14]. The next stage of CLG biosynthesis is the formation of crosslinks between tropocollagen molecules. Finally, spontaneous association occurs into microfibrils, and then into fibrils and mature collagen fibers. Type I CLG is the most common type, found in the human body [9]. It constitutes 85–90% of the organic content of bone connective tissue, skin, or other organs. It is primarily responsible for the tensile strength of tissues and bone stiffness [13,14]. It can be found in bones, which is a product of the same genes as skin type I CLG. Both of these collagens differ significantly from each other, which is the result of post-translational modifications [8,12]. Many factors can stimulate or inhibit collagen biosynthesis. This occurs at various stages of gene expression. TGF-β (transforming growth factor beta), which is one of the most important stimulating factors, influences the production of type I CLG in fibroblasts as a result of inducing the activity of the promoter of the gene encoding this protein [30]. In addition to TGF-β, the following also have a stimulating effect on the synthesis: FGF (fibroblast growth factor), EGF (epidermal growth factor), and various isoforms of PDGF (platelet-derived growth factor). The opposite effect, i.e., inhibition of CLG biosynthesis, is demonstrated by, among others, interferon α and TNF (tumor necrosis factor) [30].

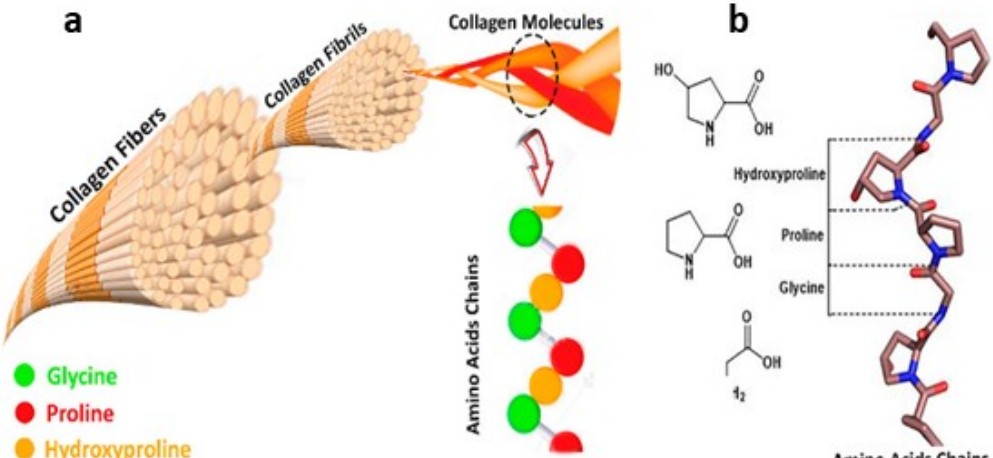

**Figure 1.** (**a**) Structure of collagen fibers; (**b**) main amino acids in collagen structure (reprinted from [28]).

### 2.2. Types of Collagen

A characteristic feature of the structure of all collagen proteins is the formation of the so-called rope, i.e., a right-handed superhelix, which is formed by winding three left-handed, single polypeptide chains around its own axis [8]. Each type of CLG has a different initiation stage in the biosynthesis process. Common steps include intracellular reactions

and triple helix formation, post-translational modifications, enzymatic glycosylation and hydroxylation of lysine, proteolytic cleavage of procollagen, and natural crosslinking [8,15]. So far, 30 different forms of collagen, which perform various functions and have different structures (Table 1), have been detected in human tissues [8]. Moreover, they differ in their location and content in the tissues (Table 1).

**Table 1.** Types of fibrillar collagens and their basic functions and occurrence in the body [23,25,31].

| Type | Structure | Localization | Function |
|---|---|---|---|
| I | Triple superhelix composed of two identical $\alpha_1$ and one $\alpha_2$ chain; a form consisting of three $\alpha_1$ chains (homotrimeric) is also found | Bones, tendons, ligaments, skin, cornea | Tensile strength of tissues and bone stiffness |
| II | Three identical $\alpha_1[III]_3$ chains—homotrimer; similar properties and size to type I, but higher content of hydroxylysine and glucose and galactose residues | The dominant component of vitreous tissue (approx. 80%), corneal epithelium, and cartilage | One of the main components of ECM (extracellular matrix), maintains chondrocyte functions, induces cell adhesion and proliferation |
| III | Three identical $\alpha_1[III]_3$ chains | Element of the dermis, liver and lung tissues, spleen and blood vessels | Gives elasticity to tissues |
| V | Three different chains—heterotrimers | Bones, skin, placenta, cornea | Initiation of collagen fibrillogenesis |
| XI | Three different chains—heterotrimers | Cartilage, intervertebral discs | Initiation of collagen fibrillogenesis |
| XXIV | The fragment of the Gly-Xaa-Yaa repeated sequence is relatively short, constitutes approximal | Bones, cornea | A marker of osteoblast differentiation and bone formation in mice |
| XXVII | Forms non-striated, thread-like structures, unlike other members of the subfamily | Cartilage | Deposited mainly in skeletal tissues at the junction of cartilage and bone |

The polymorphism of CLG may result from differences in the expression of genes encoding enzymes that are involved in the biosynthesis of these proteins [8,9,12,32]. Another hypothesis is that the diversity of these proteins is caused by changes occurring in post-translational modifications [29]. The description of the subunit composition facilitates the nomenclature based on numbering in Roman numerals (I–XXIX) [23,25,31]. Arabic numerals indicate the polypeptide chains they are composed of ($\alpha1$–$\alpha6$). The family of collagen proteins is divided into two main groups: fibrillar collagens and non-fibrillar collagens [12,14,27]. In mammals, they are encoded by 11 genes and were discovered as the first among the CLG protein family. A common feature of this group is a long central triple helix. The fibrillar molecule has a diameter of 1.5 nm and a length of 300 nm. Table 1 presents the group of fibrillar collagens, taking into account the functions, locations, and characteristics of individual types of CLG [12,14,27]. The characteristic transverse striations of CLG fibers visible under an electron microscope are caused by the aggregation of macromolecules. The vast majority of them are heterotypical. This means that they are made of more than one type of collagen (Table 2). This phenomenon is confirmed by the presence of CLG fibers in the bones and cornea, composed mainly of type I and V collagens [12,14,27].

**Table 2.** Types of non-fibrillar collagens and their location in the body [12,14,27].

| Collagen | Type | Localization |
|---|---|---|
| Basement membrane | IV | Basement membranes, capillaries |
| Forming microfibers | XVI | Bones, vessels, skin, cornea, cartilage |
| | XXVIII | Cells of the nervous system |
| | XXIX | Skin |
| Anchoring | VII | Mucosa, skin, bladder, umbilical cord, amnion |
| Forming hexagonal lattice systems | VIII | Mucosa, skin, bladder, umbilical cord, amnion |
| | X | Cartilage |
| FACITs type | IX | Cornea, vitreous body, cartilage |
| | XII | Cartilage, tendons, skin |
| | XIV | Vessels, eye, nerves, tendons, bones, skin, cartilage |
| | XVI | Heart, smooth muscles, skin, kidney |
| | XIX | Band of basement membranes in skeletal muscles, skin, kidneys, liver, placenta, spleen, prostate |
| | XX | Corneal epithelium |
| | XXI | Stomach, kidneys, vessels, heart, placenta, skeletal muscles |
| | XXII | Tissue connections |
| | XXVI | Testicles, ovaries |
| MACITs type | XII | Skeletal muscles, heart, eye, skin, endothelial cells |
| | XVII | Skin |
| | XXIII | Metastatic carcinogenic cells |
| | XXV | Eye, brain, heart, testicles |
| MULTIPLEXINs type | X | Capillaries, ovaries, heart, testicles, skin, placenta, kidneys |
| | XVIII | Kidney, lungs, liver |

However, the structure of the skin is characterized by the coexistence of types I and III, and in the cartilage, there are combinations of types II, III, IX, and XI. The composition of the fibrillar collagen group constitutes approximately 90% of all CLG proteins found in animal organisms. Fibrillar CLG types are I, II, III, V, XI, XXIV, and XXVI [8,12–14]. Non-fibrillar CLGs do not form typical fibrils. Among this group of collagen proteins, we can distinguish basement membrane, forming microfibers, anchoring, forming hexagonal network systems, the FACITs type, containing transmembrane domains—MACITs and collagens of the MULTIPLEXINs type. The affiliation of individual types of CLG with the above groups and their location in the body [14] are presented in Table 2.

*2.3. Physicochemical Characteristic of Collagen*

The molecular weight of collagen is approximately 300,000 Da, the diameter is approximately 14–15 Å (Angstrom), and the length is 2800 Å [8]. Collagen has high water-binding capacity, which makes it a good ingredient for texturizing, thickening, and creating gels [8,14]. It has properties related to its behavior on the surface: emulsion formation, foam formation, stabilization, adhesion, cohesion, film-forming properties, and a protective function for the colloid (Figure 2). Additionally, CLG is a good surfactant. It can penetrate lipid-free structures. It has good biocompatibility and low immunogenicity [6]. The proper-

ties of CLG depend on the age of the body. With age, solubility tends to decrease due to the greater crosslinking of collagen in older animals [20].

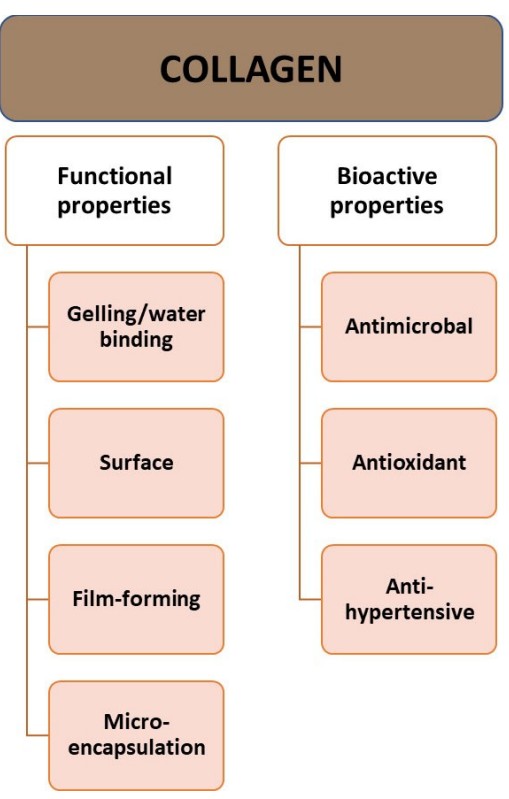

**Figure 2.** Properties of collagen.

Aging processes lead to crosslinking of the protein structure and affect its mechanical properties [15,20,33]. Mature CLG has a highly crosslinked structure and is usually insoluble in water. Water-soluble and acid-soluble CLG can only be obtained from young tissues. Age-related differences in CLG solubility have been examined based on susceptibility to pepsin digestion. Research has established that the solubility of collagen in acetic acid decreases rapidly as it matures [15]. Like other proteins, it is denatured under the influence of certain factors. If collagen is denatured by heat, the weak bonds (hydrogen, dipole–dipole, ionic bonds, or van der Waals forces) are broken, while the covalent bonds remain intact. The temperature at which thermal denaturation of CLG occurs (the so-called melting point) depends on the water content, pH of the environment, and the degree of crosslinking. The triple helix unwinds, and the chains separate. The denatured mass of tangled chains cools and absorbs all the water from the surroundings. This denatured collagen is called gelatin. Gelatin itself is a mixture of water-soluble proteins, mainly derived from collagen. It usually binds more water than CLG because it is partially degraded and more active groups are exposed to interactions with water through hydrogen bonds. Due to their structural role and compatibility with the body, collagen and gelatin are commonly used biomaterials in medicine, the pharmaceutical and cosmetics industries, with gelatin being a much cheaper material [34,35]. Collagen is a highly crosslinked material, usually insoluble in water and oils, so in the case of cosmetic preparations, it is usually hydrolyzed into smaller peptides [15,20].

*2.4. The Role of Collagen in the Body*

Collagen proteins perform various functions in the body (Figure 3). The most important of them are maintaining structural integrity, and being responsible for the processes of cell adhesion, differentiation, growth, survival, and regeneration [12,30,36]. CLG is found

in various tissues of the body (Tables 1 and 2), and its structure varies depending on its location and function.

## COLLAGEN PEPTIDE

### Skin system

- Inhibits skin aging
- Improves skin properties (hydration, elasticity, and dermal collagen density)
- Inhibits wound inflammation
- Promotes epithelial cell formation, tissue regeneration, growth factor expression

### Cardiovascular system

- Reduces vascular pressure
- Reduces blood glucose, triglyceride, cholesterol
- Regulates insulin secretion
- Promotes fat metabolism

### Nervous system

- Relieves age-related learning
- Prevents cognitive function, anxiety-like behavior, and stress response defects

### Skeletal and muscular system

- Relieves osteoporosis
- Increases bone density and bone strength
- Improves body composition and local muscle strength

### Gastrointestinal system

- Protects intestinal epithelial function
- Affects nutrition metabolism
- Changes intestinal flora

### Immune system

- Inhibits tumor cell migration and proliferation
- Enhances cellular immunity and humoral immunity
- Suppresses allergic reactions

**Figure 3.** Main role of collagen in the body.

The basic task of collagen is to connect cells, which is why it is a building block of most organs, especially skin, bones, teeth, cartilage, blood vessels, and the cornea of the eye [28]. At the same time, it protects internal organs such as the kidneys, stomach, and liver, creating a flexible scaffolding around them. It also takes part in regenerative processes and ensures proper hydration of the skin thanks to its ability to bind water [23,25,37]. In the immune system, it prevents the entry of pathogenic microorganisms and toxic substances [30]. It ensures the continuity of cell renewal processes in the skin and maintains the appropriate level of hydration, which affects its elasticity, appearance, and condition [33,38,39]. It accelerates wound healing, creates scars, and promotes the reconstruction of connective tissue [40,41]. Collagen increases the absorption of minerals and increases bone density. It stimulates the activity of cartilage cells and supports protective processes within cartilage tissue, providing cartilage with the appropriate shape and resistance to stretching [38]. It is responsible for the production of synovial fluid and the condition of cartilage. Moreover, it reduces the activity of enzymes responsible for causing inflammation and rheumatic pain [30,42]. CLG fibers can also be carriers of some drugs, including interferon. It provides essential amino acids that nourish hair bulbs and ensure their proper growth [34]. Its proteins also play an important role in the functioning of the circulatory system because they are a component of blood vessel walls. The functioning of the circulatory system directly depends on the composition and structure of the vessel walls [6,7]. Collagen, as a fibrillar support protein, interacts with the structures of the extracellular matrix of vessel walls, which gives them appropriate elasticity and mechanical strength. CLG reorganization can impair the effective function of veins and arteries in transporting blood. It also significantly influences the development of circulatory system diseases [23,25]. Collagens are the largest group of proteins that make up the blood vessels of the human body; types I and III constitute approximately 90% of all CLG in the vessels. They are the main supporting proteins of vessels—both healthy and damaged ones. However, among the non-fibrillar collagens that support the functioning of vessels, we can distinguish type VI CLG, which forms microfibers, and collagens that combine with fibrillar CLG—types

XII and XIV [13,26]. In the basement membrane of the inner layer of the vessel, there is also a network of CLG molecules of type IV, anchoring of type VII, and CLG of types XV, XVIII, and XIX, which act as connectors between cells and the basement membrane. Due to the high stiffness of their fibers, their main mechanical role is to limit the expansion of blood vessels under the influence of the pressure of blood flowing through them [25–27]. The number of bonds between collagen molecules affects the quality of the mechanical functions of the vessel walls, as well as the organization of crosslinked collagen fibers within the tissue and their interaction with other matrix components [14,26].

### 2.5. Effects of Collagen Deficiency in the Body

In a healthy, young body, collagen is regularly replaced—about 3 kg per year—and systematically rebuilt, and over time, the ability to regenerate CLG fibers disappears. The level in the body begins to decrease from the age of 25, its sharp decline occurs after the age of 50, and after the age of 60, this protein ceases to be synthesized by the body. However, the reduction in endogenous collagen biosynthesis is not limited to adulthood [8,14,27]. A decrease (Figure 4) in its concentration in the body can also be observed in young people, which is genetically and hormonally determined. This process is also favored by stress and other external factors, and the CLG biosynthesis process is also disrupted during menopause.

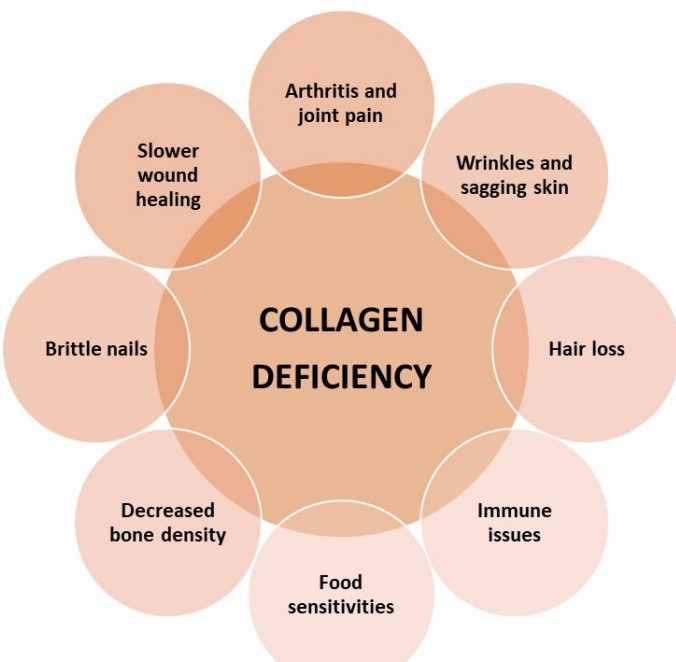

**Figure 4.** The main effects of collagen deficiency in the body.

Then, the level of estrogens decreases, which affects the synthesis of this protein and changes its properties. CLG fibers become less elastic and thinner [7,16,23,27,36,43]. Changes in the structure can occur under the influence of various external factors, such as mechanical loads, hormonal changes, or diseases [8,32,44,45]. Increased physical activity and practicing extreme sports contribute to the increased destruction of collagen fibers and disturbances in their synthesis. Accelerated fiber breakdown is also caused by excessive exposure to the sun, too high and low temperatures, and some compounds contained in cosmetics [10,46]. These changes may result (Figure 4) in problems with movement, spine and joint pain. With age, the concentration of vitamins A, C, and E, and minerals (copper), which are responsible for the natural renewal of collagen, decreases. Disturbances in the synthesis of CLG protein and its transformation in bone tissue may cause bone decalcification, brittleness, and susceptibility to fractures. The most noticeable symptom is

the loss of skin firmness and general deterioration of its condition. Hair becomes brittle and falls out excessively, which may lead to premature baldness [17,37,47,48].

## 3. The Importance of Collagen in the Skin

The main role in the structure of the dermis is played by collagen fibers; they constitute 80% of the connective tissue, 72% of the dry matter of the skin, and 30% of all proteins that make up the body. In addition, they can be found in the underlying connective tissue, cartilage, tendons, and muscles. They exist as network structures by crossing and intertwining with each other (Figure 5) [45]. Despite their slight extensibility, they are characterized by significant tearing strength. CLG fibers cut the gel of the proteoglycan matrix and thus provide it with tensile strength. They are constantly being rebuilt—new CLG is created in place of damaged, old protein [33,49]. However, it can be destroyed, e.g., by UV rays (as a result of photocrosslinking) [46,50], sugar molecules (as a result of glycation) [44,51,52], and free radicals (released as a result of diseases, smoking, stress, or practicing high-level sports) [33,43,49,53,54]. There is a natural mechanism in the skin that regulates the production of collagen.

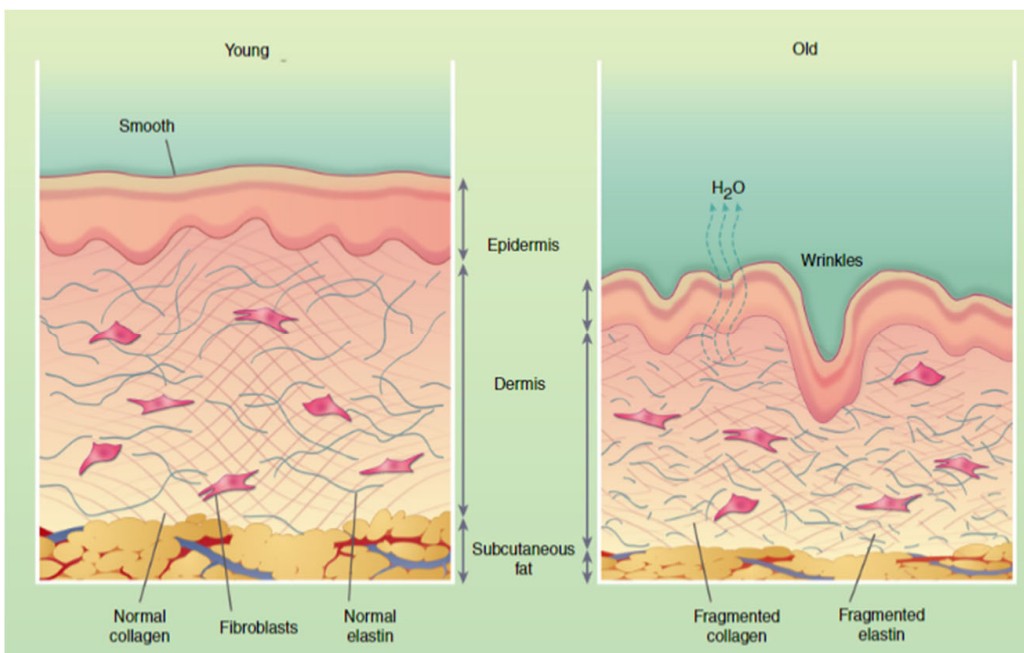

**Figure 5.** Comparison young and aged skin (reprinted with permeation from [45]).

Enzymes—metalloproteinases—play an important role in this process, e.g., a collagen molecule damaged by UV rays is cut into pieces by MMP 1 (matrix metalloproteinase I) [11,55]. Such fragments created from cut peptides (e.g., tripeptides) become feedback for fibroblasts, which will be able to produce new, properly constructed collagen. Cosmetic procedures are largely based on this knowledge, the mechanism of which is to destroy old CLG fibers, then produce signal peptides from them, and then stimulate fibroblasts to produce new CLG [53]. Between the epidermis and the dermis, there is a basement membrane, which is a thin layer of intercellular substance on which epithelial cells rest. Cells are connected to the membrane through hemidesmosomes and focal attachments [43,53,54]. The epidermis is placed on the basement membrane to provide proper support and proper proliferation of its cells. This also allows for it to adhere properly to the dermis. Moreover, the basement membrane participates in the exchange of nutrients between blood vessels and avascular epithelium and in the differentiation of epithelial cells [45]. The basement membrane consists of i. lamina lucidum, which lies below the epithelial cells; it contains characteristic glycoproteins such as fibronectin, laminin, and entactin, which ensure adhesion, i.e., adhesion of the basement membrane to epithelial cells and connective tissue

elements; ii. lamina densa, made of type IV collagen, which creates a multilayer network that scaffolds the basement membrane; iv. reticular lamina, composed of reticulin fibers consisting of type III collagen [13,27,45]. The building block of anchoring fibers that are involved in attaching the basement membrane to the connective tissue substrate is type VII collagen. The lamina lucidum and the lamina densa form the basal lamina, which, apart from being present in the epithelia, also covers fat and muscle cells. Collagen proteins are characterized by high resistance to physical, chemical, and, above all, mechanical factors [34,36,47,56]. Thanks to this, CLG can bond structural elements at the cellular level. By bonding individual tissue elements together, it ensures their complete integrity. It also fulfills this function at higher levels of cellular organization, e.g., in the skin—it ensures the integrity of the dermis, consisting of connective tissue, with the epidermis, which is an epithelial tissue [45]. Because CLG fibers form an ordered spatial network, other elements of the extracellular matrix can attach to them via receptors. When a mechanical stimulus is activated, the network deforms and then returns to its initial state when the factor ceases to be active [45]. Thanks to this ability to deform the collagen network, the elements attached to the fibers or located close to them will not be damaged under pressure and can continue to perform their functions properly [23]. Another important feature of CLG is the ability to bind water. For protein fibers to retain their properties and structure, adequate hydration is necessary [56]. The skin is then appropriately tense and elastic and can act as a protective barrier and be resistant to mechanical factors [23]. Since most biochemical reactions take place in an aqueous environment, high water content is essential for the proper functioning of the epidermis and dermis. In dry skin, a decrease in metabolism is observed. Research suggests that collagen molecules also play a role in the process of cell proliferation, thanks to their ability to bind cytokines, and the stimulation of cell division in the skin is extremely important, especially in the process of regeneration of skin defects [30,37,48]. According to research, amino acids derived from the breakdown of fish collagen (glycine and hydroxyproline) [3,19], which was applied to the skin, have a possible impact on the amount of cytokines produced by fibroblasts and keratinocytes. Transdermal fish CLG stimulates the synthesis and secretion of the following cytokines: FGF and TGF [6]. TGF-β in skin fibroblasts is an important regulator of CLG homeostasis, stimulates procollagens I and III, and also reduces MMP-1 transcription [17].

### 3.1. The Influence of External Factors on Collagen Fibers

Collagen constantly changes its structure as a result of external factors (Figure 6) [9]. Collagen structures are damaged by factors such as UV radiation, hormonal changes, mechanical loads, vascular changes, and inflammation [15,35,37].

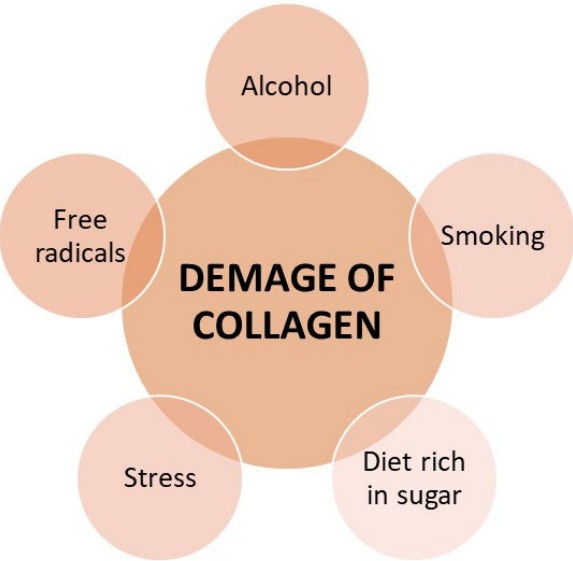

**Figure 6.** Factors influencing the structure of collagen.

Damaged collagen is cut by collagenases (matrix metalloproteinases) so that it can be replaced by new, properly crosslinked collagen. Among the factors that damage collagen proteins, we can also distinguish free radicals generated under the influence of stress [36], drinking alcohol or smoking cigarettes, and an incorrect diet rich in sugar causing glycation of collagen and low in, e.g., flavonoids, i.e., anti-inflammatory factors. Toxic compounds may be factors that block collagen synthesis, e.g., heavy metals such as mercury, lead or cadmium [8,14,34,36,47,53,54,57].

### 3.1.1. Collagen Glycation

As a result of the glycation process, the so-called AGE (advanced glycation end-product)—i.e., products of advanced glycation—compounds have the ability to create crosslinks in proteins [44,51,52,58]. Glycation (Figure 7) is a non-enzymatic reaction between reducing sugars and proteins, lipids, or nucleic acids. Enzymatic glycosylation of proteins in the body is a deliberate process and occurs in a specific place, while glycation, i.e., non-enzymatic glycosylation, is spontaneous [44,58]. Its severity depends on the content of simple sugars in the body, e.g., glucose. AGEs, both directly and indirectly, cause the destruction of many cellular structures, including by inducing oxidative stress [59,60].

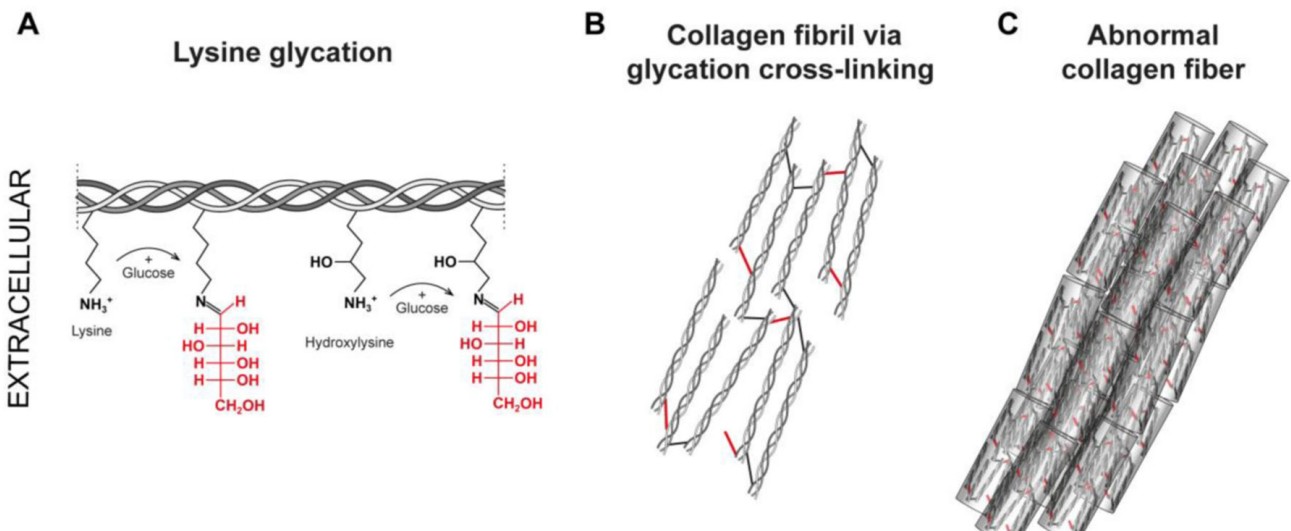

**Figure 7.** Steps of glycation of collagen fibers, (**A**) Lysine glycation (red font—reduced glucose attached to lysine molecule and forming a Schiff base); (**B**) Collagen fibril via glycation cross-linking (red lines—bonds formed during the glycation process); (**C**) Abnormal collagen fiber (red lines—bonds formed between fibers) (reprinted from [58]).

Protein glycation involves the reaction of reducing sugars (glucose, fructose, glucose-6-phosphate) with the amino groups of proteins and takes place in stages. Initially, the free amino group (mainly arginine or lysine) interacts with the carbonyl groups of sugars [44]. As a result of this action, a Schiff's base is formed, which in the next stage is rearranged to enaminol. [44,51,52]. The next stage, i.e., the transformation of the resulting product into ketolamine, is irreversible. Enaminols and ketolamines are early products of glycation, while final glycation products are formed as a result of the Maillard reaction. As for collagen molecules, asparagine, polyproline, or glutamine take part in the formation of the Schiff's base. They react with the carbonyl group of sugar—usually the aldehyde group of glucose and galactose or the ketone group of fructose [51]. In living organisms, the glycation process occurs slowly, over many weeks, because a small part of glucose exists in a form containing a free aldehyde group (open form), which could react with the amino groups of proteins [52]. Therefore, this process mainly involves proteins with a long half-life, such as collagen. They then become resistant to proteolysis, which accelerates their aging process [36]. As a result of the glycation of type IV collagen, which is responsible for the

structure of the basement membrane, Its turnover process is significantly slowed down. However, glycation of type I collagen and proteoglycan causes stiffening of their fibers and a reduction in their elasticity [31,44]. The AGE formation process is catalyzed by copper and zinc and inhibited by, e.g., ascorbate (reducing agent). Glycation of proteins is promoted by, among others, insulin resistance, hyperglycemia, as well as low levels of antioxidant factors such as vitamin E, vitamin A, and selenium. The final glycation products may not only be of endogenous origin—but their source may also be food products in which AGEs are formed as a result of heat treatment [44,51,52].

### 3.1.2. The Effect of UV Radiation on Collagen Fibers

Ultraviolet radiation is characterized by high biological activity. Radiation absorbed by a given substance may cause changes in its physicochemical properties, e.g., ionization or heating [46]. Moreover, it may cause the initiation of photochemical reactions such as oxidation, reduction, decomposition, or polymerization [33,36,53,54]. Ultraviolet radiation leads to the thickening of the stratum corneum (SC) as a result of increased cell division in its creative layer [10,46]. The thicker the SC, the more radiation it can absorb. Direct contact of the skin with UV rays does not benefit the tissues, the main structures exposed are proteins [55]. About 50% of UVA radiation reaches the dermis [46,50]. However, before damage occurs, the body activates various defense mechanisms. By stimulating individual membrane receptors, information about the radiation reaching the skin is transmitted to the cell nucleus, where the transcription of various genes is activated. Genes encoding metalloproteinases are activated, which attack collagen, hyaluronic acid, and elastin molecules and form the stroma of the dermis [49]. The c-Jun transcription factor is also activated, which is intended to inhibit CLG synthesis until the cell repairs any damage. The human body can lose up to 20% of normal skin collagen as a result of careless exposure to the sun in one day [11,55].

UVB radiation stimulates the expression of genes responsible for the apoptosis of fibroblasts, melanocytes, and keratinocytes [53]. In tissue culture studies, fibroblasts and keratinocytes collected from tanned skin have a shorter lifespan compared to cells from untanned areas. As a result of radiation, the process of elastosis also takes place, in which amorphous, thickened elastin fibers move toward the upper layers of the dermis, where they replace collagen fibers [46]. UV rays cause photocrosslinking and photodegradation of collagen. As a result of photocrosslinking, abnormal connections are created between fibers, which changes their mechanical properties [11,16,46,50,55]. Free radicals produced under the influence of radiation, which are capable of further reactions, also have a harmful effect (Figure 8) [49]. As a result of irradiation of the skin surface, the energy of UV rays is easily absorbed by cellular chromophores such as NADH2, NADPH, riboflavin, tryptophan, and trans-urocanic acid [49].

The stimulation of chromophores occurs with the participation of molecular oxygen, which results in the formation of oxidation products and oxygen radicals, including the highly reactive hydroxyl radical HO* [10,46]. Free radicals stimulate the synthesis of collagenases that maintain CLG damaged by photocrosslinking [10,50]. As a result of the reaction of free radicals to CLG fibers, tissue stiffness and density increase, and skin pigmentations and wrinkles form [11,50,55]. However, prolonged exposure to UV radiation on the skin may lead to the development of rheumatic diseases, inhibition of wound healing processes, and the development of cancer [46,50]. In studies on the effect of UV radiation on collagen isolated from fish scales, conformational changes in the CLG molecule and water loss were observed [59,60].

Initially, absorbed UV rays cause the loosening of the protein structure and then its degradation. Under the influence of radiation, hydrogen bonds that stabilize the triple helix are broken, which leads to changes in conformation and the transition of collagen molecules from a helical structure to a coil structure. Photodegradation results from the disruption of peptide bonds as well as the formation of new products: phenylalanine and tyrosine [33,60]. Moreover, as a result of radiation exposure, an initial increase and then a

rapid decrease in the optical activity and viscosity of CLG were observed. In the study of mouse skin collagen, crosslinking of type I and IV was observed as a result of radiation with a wavelength of 290–320 nm [11]. Type IV collagen produced 3,4-dihydroxyphenylalanine (DOPA), which is involved in skin aging, while the analysis of the amino acid composition showed that the amount of aromatic amino acids decreases under the influence of UV rays [47]. Taking into account previous research on the photostability of collagen, it can be concluded that UV radiation is harmful not only to the tissues in which protein occurs but also to preparations containing this ingredient, e.g., cosmetic products. To sum up, UV radiation causes a deficit in collagen as a result of shifting its homeostasis from production to degradation [11,46,53,55,59–61].

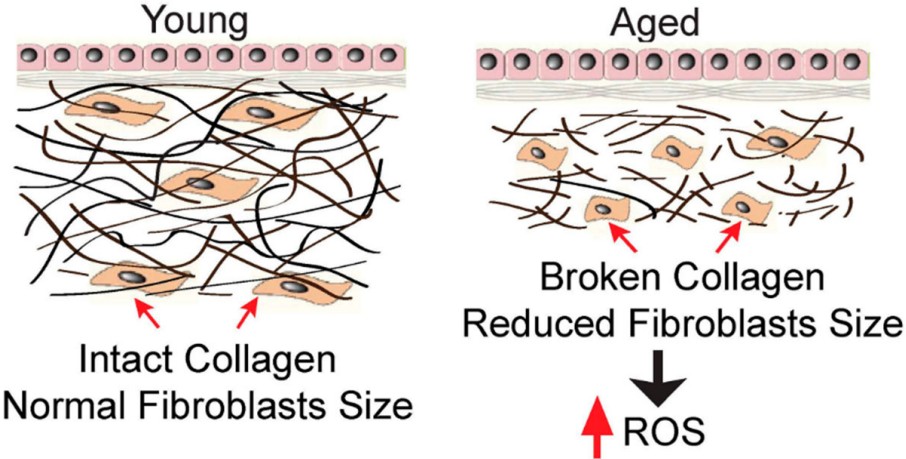

**Figure 8.** Comparison of fibroblasts and collagen fibrils in young and aged skin (reprinted from [49]). Reduced fibroblasts size stimulates intracellular ROS generation. In the young human skin dermis, intact collagen fibrils interact with cells to maintain normal cell spreading and size (**left**). In contrast, in the aged human skin dermis (**right**), broken collagen fibrils are unable to support normal cell spreading, and this causes reduced cell size. One of the prominent features of the collapsed cells is the increase in intracellular ROS generation.

## 4. Characteristics of Cosmetic Products Containing Collagen

CLG is obtained from many sources; the main one is beef, especially from cow bones and skin. Due to the occurrence of cow diseases that threaten human life, such as BSE (bovine spongiform encephalopathy), TSE (transmissible spongiform encephalopathies), FMD (fibromuscular dysplasia), and mad cow disease in particular, scientists are looking for an alternative, safer source of collagen [28,40,62–67]. One of the main disadvantages of bovine CLG is that approximately 3% of the total human population is allergic to it, making it difficult to use. Beef Achilles' tendon is used to obtain type I CLG. Type IV is obtained from the placental villi, while type II is obtained from the nose or articular cartilage. Cattle are used at various stages of their development. The dermis of the fetus is used to strengthen tendons and improve wound healing, while the dermis of newborn cattle is used to treat hernias and is also used in plastic and reconstructive surgery [29,40,41]. The pericardium of adult cattle has been used in the treatment of hernias and muscle strengthening [29]. Porcine collagen is widely used for industrial purposes; pig bones and skin are used. Pork CLG is very similar to that of humans, but it does not cause as many allergic reactions. As with bovine CLG, there is a risk of zoonotic diseases. The dermis of adult pigs and the mucosa of the small intestine are used to strengthen tendons, treat hernias, support wound healing, and in plastic and reconstructive surgery [12,13,29]. Currently, the safest source of collagen is considered to be marine [3,6,18,19,28,67].

Extracting animal collagen is complex, expensive, and time consuming. Due to concerns about the occurrence of defense reactions and the occurrence of diseases among land animals, marine sources began to be investigated (Figure 9) [62]. Marine CLG has many advantages over that received from land animals: mainly, lack of diseases, environmental

friendliness, better absorption due to lower body temperature (than terrestrial animals), greater absorption due to low molecular weight, minimal presence of biological contaminants and toxic substances (practically negligible), lower immunogenicity, and metabolic compatibility [29,62,63].

Marine sources include the use of fish, starfish, jellyfish, sponges, sea urchins, octopus, squid, cuttlefish, anemones, and shrimp [6,18]. Freshwater and saltwater fish are used to obtain fish collagen, specifically their skin, bones, fins, and scales. These fish parts are considered waste during fish processing, so using them to obtain CLG reduces environmental pollution [66–68]. Type I CLG is obtained mainly from the skins of Atlantic cod, silver carp, Japanese sea bass, chub mackerel, and sole and from the bones of carp, yellow sea bream, and Japanese sea bass [29]. Other animal sources include chicken, kangaroo tails, rat tail tendons, duck legs, horse tendons, alligator skin and bones, sheep skin, frog skin, and sometimes even human skin [62]. Recombinant human collagen is used due to its lower immunogenicity compared to other sources [63,69]. The pericardium of adult horses is used to strengthen tendons, treat and heal wounds, and treat hernias. Collagen types I and II are isolated, among others, from horse skins and their joint cartilage. Types I, II, III, and V are obtained from chicken necks, with type I being dominant [70]. Chicken feet are also a rich source of CLG. Type I is also obtained from the oviducts of bullfrogs [29].

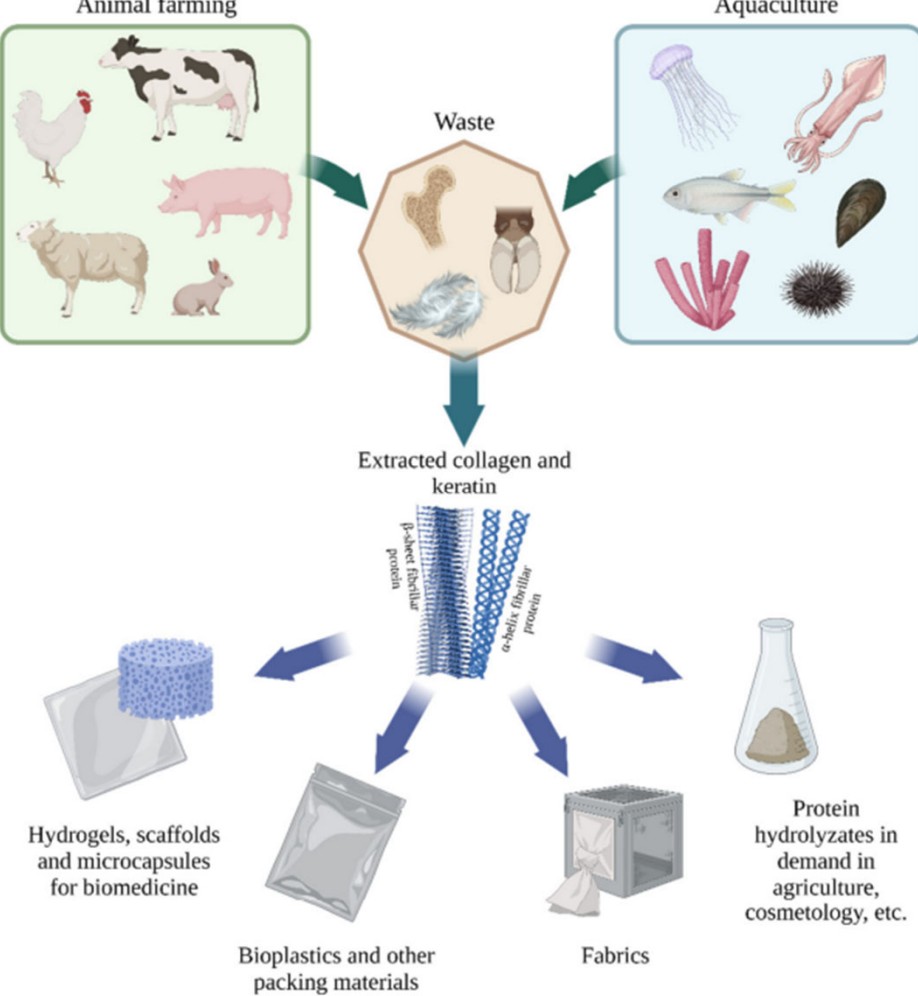

**Figure 9.** Sources and application of collagen (Reprinted from [62]).

### 4.1. Sources of Collagen Used in Cosmetics

Since collagen acts as a natural humectant and has moisturizing properties, it is one of the basic ingredients of many cosmetic products [69,71,72]. The cosmetics industry is

constantly developing and looking for innovative, effective products, which makes collagen sources the subject of many studies. CLG used in cosmetics comes from many sources: from bovine through pork to marine [1,4,73]. However, as previously mentioned, pork and bovine collagen is associated with the risk of zoonotic diseases [15,19,20].

The research contributed to understanding the structural and physicochemical features of CLG of marine origin, which constitutes a significant disjunction with animal sources. CLG derived from jellyfish, more specifically from the species *R. pulmo*, has similar biological activity to human collagen type I [6,7,18,19]. Studies have shown that certain human receptors can identify collagen derived from jellyfish. It follows that there is a similar response regarding cell adhesion, proliferation, and migration to jellyfish CLG [15]. The protein amount in jellyfish is approximately 60%. Test results showed that the amino acid level was similar to that of vertebrate CLG, but much lower hydroxyproline content was observed. This indicates relatively low denaturation temperatures, ranging from 26 to 29.9 °C [15,20]. Another source of CLG for cosmetic purposes is cuttlefish skin [28]. The Sepia lycidas species contains between 2 and 35% of collagen in freeze-dried dry matter. However, Argentine squid skin contains approximately 35.6% CLG. Among marine sources, fish are the most valuable source of CLG. The main reason is that 75% of a fish's weight is collagen. Fish skin is mainly used to obtain type I CLG [28]. Type II is found in cartilage. According to research, the highest collagen content was observed in silver carp, brown toads, cod, and tilapia [4,15,18,28,66–68].

As mentioned, marine collagen, depending on its source, can be divided into two categories—isolated from invertebrate or vertebrate animals [28,61,65,67,68,74]. Type IV collagen received from sponges is used in cosmetics for dry skin and type I CLG from salmon skin has moisturizing properties in cosmetics. For the extracted peptide to be suitable for use in cosmetics, it must be properly isolated and purified. Due to the presence of strong crosslinks in the triple helix structure of CLG, it is poorly soluble in cold water. Although solubility is improved by heating, chemical treatment of the animal tissue is necessary, so dilute acids and bases are used to break the links. The acid most often used to extract CLG from tissues is dilute acetic acid, as well as lactic and citric acid [6]. As a result of research on acid extraction of CLG, it was observed that CLG isolated from salmon skin was more soluble than that isolated from cod skin [3]. However, CLG also has strong intermolecular covalent bonds in the telopeptide regions, which cannot be cleaved by acetic acid alone—in this case, enzymes are also used, e.g., pepsin, trypsin, papain, collagenase [6]. Thanks to their use, the solubility of collagen and the efficiency of its extraction increase. The most commonly used enzyme is pepsin [6,65]. Many cosmetic preparations contain CLG (Figure 10) as a main component due to its benefits of being a natural humectant and having moisturizing properties. Preparations based on marine collagen are promising alternatives due to the continuous development and search for innovative and effective products. They will differ in composition and properties depending on their origin [1,56,57,65,67,74]. High molecular weight proteins such as CLG are unable to be absorbed by the stratum corneum, so they remain on the surface, preventing water loss—they maintain proper hydration of the skin and protect against microorganisms in the case of injured tissue.

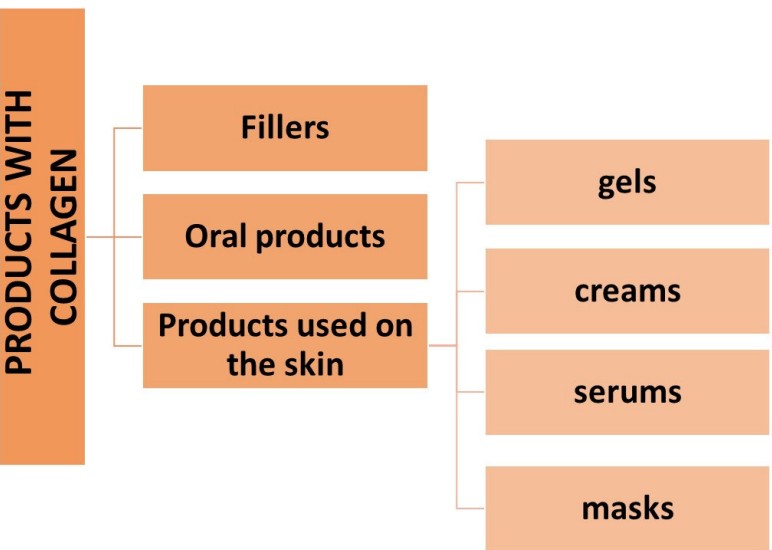

**Figure 10.** Types of products with collagen.

*4.2. Collagen Fillers*

The use of collagen fillers is an increasingly developing method in minimally invasive aesthetic medicine and involves leveling the skin surface, i.e., "smoothing" wrinkles and places with, for example, atrophic scars [75,76]. They are also used to shape lips or face contours. Fillers are substances that can fill soft tissue defects permanently or temporarily and are injected intradermally [4,16,72]. We can distinguish synthetic and natural fillers, which are obtained from animal tissues (xenogeneic), human tissues (allogeneic), and our tissues (autogenic) [15,16,20,75,76]. Collagen fillers belong to the group of the oldest tissue fillers; they are divided into non-absorbable (permanent), and absorbable (temporary). Their feature is full biodegradability; they are also relatively cheap and easy to produce, non-carcinogenic, non-teratogenic, and non-toxic, and provide repeatable results, which is why they are gaining popularity [15,16]. Their significant disadvantage is the occurrence of side effects of varying intensity, depending on the preparation used, such as erythema, swelling, itching, infections, or discoloration [72,75]. As a result of the search for collagen fillers with lower immunogenicity, preparations such as Cosmoderm® and Cosmoplast® (INAMED Aesthetics, Santa Barbara, CA, USA) were introduced to the market. Both of these products are obtained by culturing human fibroblasts and contain mainly type I collagen (as much as 93%) and type III CLG (7%), as well as lidocaine [72,77]. They belong to allogeneic fillers and are obtained from the tissues of deceased people after they have been previously tested for the presence of teratogenic and infectious factors. Cosmoderm® specializes in the correction of more superficial wrinkles, while Cosmoplast® is used for deeper wrinkles, the nose, and lifting the corners of the mouth, as well as for correcting the contours of the lips [76]. When using these preparations, allergy tests are not required, and the effects last from 3 to 7 months (Cosmoderm® 3–4 months, Cosmoplast® 3–7 months) [72]. One other collagen preparation is Isolagen® (FibroCell Science, Exton, PA, USA), which is obtained from the patient's skin to create a suspension of live fibroblast cultures and extracellular matrix [75]. The effect of this preparation appears with a delay because it is related to stimulating the remodeling of existing tissues and the synthesis of new components of the extracellular matrix. After the period necessary to perform cell culture (about 6 weeks), it is administered by injection, and it is recommended to perform an allergy test beforehand [72]. A representative of the xenogeneic filler is Artefill® (Artes Medica, San Diego, CA, USA) [75], containing bovine collagen, which is also the filler with the longest-lasting effect. Bovine CLG biodegrades and is replaced by the body's collagen after 1–3 months. Collagen fillers are also used to fill the lips, and the collagen concentration in such preparations is 3.5–6.5% [15]. To sum up, xenogeneic fillers have a wide range

of applications in medicine, e.g., in surgical sutures, but the factor limiting their use is immunogenicity [75,76]. However, autogenous and allogeneic fillers are characterized by high therapy costs, and if the products are not thoroughly cleaned, there is a risk of transmitting pathogenic prions or viruses to the patient [72].

### 4.3. Oral Products with Collagen and Its Hydrolysates and Their Effects

A typical CLG hydrolysate consists of peptides of various lengths depending on the source and is characterized by a special amino acid composition [17]. So far, oral preparations with collagen or its hydrolysates were mainly intended for people struggling with connective tissue diseases, mainly cartilage and joint changes [38]. However, there are more and more dietary supplements with collagen appearing on the market, the effect of which is to improve the appearance of the skin and delay its aging. Such preparations with aesthetic indications have been popular for a long time, e.g., in Japan, where they constitute a significant category of nutraceuticals, while in Europe they appeared later [72]. Hydrolyzed CLG works in the dermis [33] in two ways (Figure 11): first, free amino acids provide the building blocks for the formation of collagen and elastin fibers, and second, collagen oligopeptides act as ligands—they bind to receptors on the fibroblast membrane and stimulate the production of new CLG, elastin, and hyaluronic acid [33,63]. In the literature, there are descriptions of studies on the impact of oral preparations with CLG on the body [38,42,78] in which an increase in collagen synthesis by fibroblasts (thanks to the amino acids contained in CLG) increased proliferation of skin cells and increased CLG synthesis due to the presence of hydroxyproline. Taking collagen hydrolysates resulted in an increase in skin hydration [22,72].

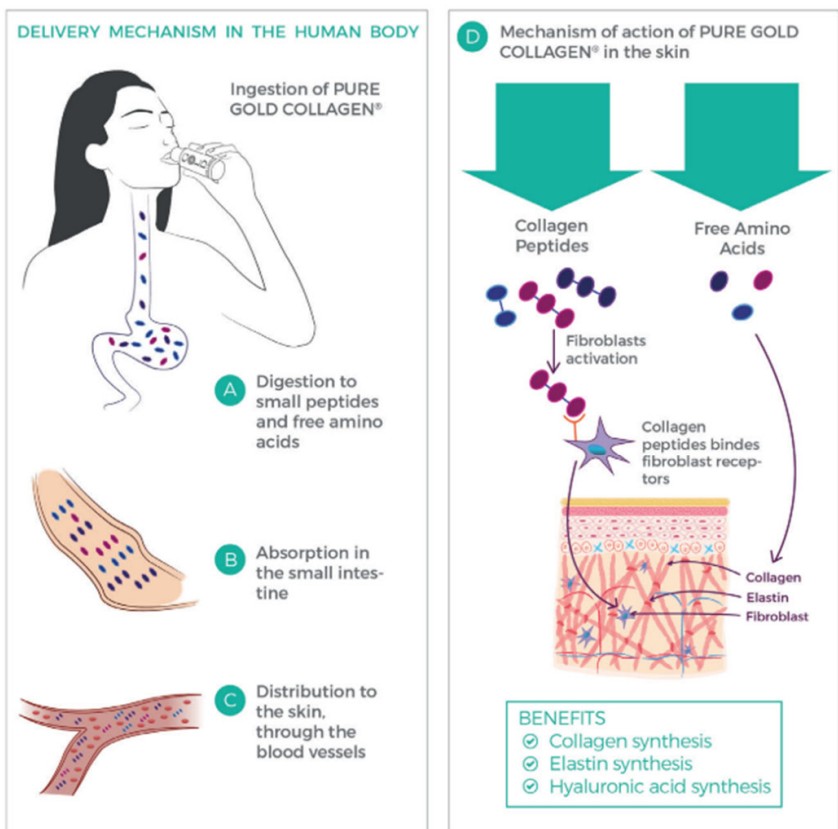

**Figure 11.** Delivery mechanism of collagen in human body (**A–C**); effect of action of collagen in the body (**D**) (reprinted from [33]).

Numerous studies have shown that as a result of consuming hydrolyzed collagen, peptides are produced in the bloodstream (after oral consumption, CLG hydrolysate is degraded to dipeptides and tripeptides [15]). After this, they have chemotactic properties for skin fibroblasts, supporting their reconstruction processes [20]. Other studies have shown that oral consumption of a preparation containing bioactive CLG peptide (BCP) by a group of women aged 45–65 years resulted in a significant reduction in, among others, wrinkle depth—a difference of 7.2% compared to the placebo group after eight weeks of use, this difference increased to 20.1%, and four weeks after taking the last dose, the wrinkle depth reduction effect remained at the level of 11.5% [79]. After eight weeks of BCP use, the content of tropocollagen type I was as much as 65% higher than in the placebo group, which could have contributed to greater synthesis of natural CLG and, consequently, to increased skin elasticity and better hydration [15]. In the case of hydrolyzed fish collagen, which has lower stability compared to animal one due to a smaller amount of hydroxyproline and a larger amount of amino acids such as serine, methionine, and threonine, liposomes have been used to increase the bioactivity of CLG [20]. Studies conducted on rats have shown that daily supplementation with bovine CLG hydrolysates has a positive effect on extracellular matrix proteins—the amount of type I and IV CLG in skin samples increased [78], while number of enzymes responsible for its degradation decreased, including type II collagenase. Based on such studies, it can be concluded that supplementation with collagen hydrolysates may delay the formation of changes in the extracellular matrix that occur with age, and the mitigating effect of these preparations on oxidative stress has also been proven [72,77,80]. When supplementing with oral products with CLG, it is worth ensuring the appropriate level of vitamin C, because it plays an important role in the synthesis of collagen fibers [33]. There are a lot of websites that recommend dietary supplements in sachets with CLG. Usually, they contain 5000 mg of collagen with a molecular weight of 3000 Da in one sachet, which, according to the manufacturer's description, is characterized by high absorption and provides protection against oxidative stress [79]. The CLG in the preparation is of marine origin, more specifically from cod fish, and is intended to restore the skin's density, elasticity, and hydration. According to the opinions of customers who used such products regularly for a month, the level of skin hydration and its density improves [33,38].

As previously mentioned, oral supplementation with collagen preparations is currently very popular. Table 3 summarizes information presented in various publications about research on these effects.

An example of a study prepared to be conducted on the effect of collagen supplementation on skin hydration, elasticity, smoothness, and density was proposed by Bolke and co-workers [17]. The study aimed to investigate the effect of the drinking nutraceutical ELASTEN® (QUIRIS Healthcare, Gütersloh, Germany) on aging and skin health. The ampoules were supposed to provide a mixture of 2.5 g of CLG peptides, acerola extract, zinc, biotin, vitamin C, and a native vitamin E complex. The study was conducted on a group of 72 healthy women aged 35 and over who took a supplement or a placebo for 12 weeks [17]. Then, the skin properties were assessed using a corneometer (assessment of skin hydration), a cutometer (assessment of skin elasticity), a silicone skin replica with in vivo optical measurements with a 3D phase shift (assessment of skin roughness), and a skin ultrasound (assessment of skin density) [17]. The tested ELASTEN® collagen complex contained short-chain oligopeptides containing between 5 and 26 amino acids, which largely overlap with the amino acid sequences found in collagen proteins of human skin. They are obtained as a result of the enzymatic hydrolysis of natural CLG, and after consumption, they are further metabolized in the digestive tract to bioactive di- and tripeptides. Then, they reach the bloodstream and the skin, where they accumulate to form a CLG biomatrix [17].

**Table 3.** The impact of collagen supplementation on individual systems and organs of the human body [17,38,47,63,81,82].

| Place of Action | Effect of Supplementation |
|---|---|
| Immune system | Inhibition of inflammatory processes resulting from autoimmunity. |
| Hormone system | Amino acids included in collagen may be involved in the production of hormones and neurotransmitters, e.g., GABA. |
| Cartilage and skeletal system | Increased production of collagen in cartilage—improved resistance to mechanical loads, reduction in joint pain and joint inflammation, stimulation of the production of synovial fluid. |
| Skin | Bioactive short peptides (dipeptide glycine—hydroxyproline) send signals to fibroblasts, stimulating the collagen production process, reducing wrinkles, improving the smoothness, tension, and elasticity of the skin, reducing cellulite and stretch marks by improving the elasticity and cohesion of the skin. |
| Hair, nails | Preventing graying and hair loss and brittle nails. |
| Liver | Assistance in the removal of toxic substances and alcohol metabolites. |
| Muscle | Increase in muscle mass, increased energy efficiency of the body. |
| Circulatory system | Amino acids contained in collagen, such as proline and lysine, prevent the accumulation of fats in the walls of arteries—protection of the circulatory system. |
| Digestive system | The amino acids proline and glycine seal the walls of the small intestine; collagen helps absorb water in the intestines. |

Studies have shown (Figure 12) that age-dependent reductions in CLG synthesis can be reversed by oral administration of specific bioactive CLG peptides. Objective dermatological measurements such as corneometry and cutometry confirmed that oral use of collagen peptides along with other ingredients significantly improves skin hydration, density, elasticity, and roughness after three months of use [17]. In addition, study participants, in their subjective assessments, concluded that the appearance of their skin had improved significantly. The CLG supplement did not cause any side effects, was well tolerated, and was safe to use. Since CLG peptides were taken orally, the effects reached deeper layers of the skin, permanently improving its physiology and appearance [17,33]. The subject of the next study was to compare the consequences of using oral collagen with the use of topical collagen preparations in reducing or delaying skin aging. Evidence from peer-reviewed studies suggests that both oral and topical CLG preparations improve skin hydration and elasticity, and skin hydration improves when administered orally. Additionally, CLG has been proven to reduce wrinkles and skin roughness, and existing research has not shown any side effects from the use of CLG supplements [38,42,83]. Among the publications was a study conducted in Japan in which collagen peptides were administered orally to patients with aging, wrinkled skin [83]. The study involved women over 40 years of age who used 10 g of collagen for 56 consecutive days or a placebo. The authors observed a statistical difference in skin hydration during the entire experiment compared to the placebo group. A device using bioelectrical impedance analysis was used to test skin hydration—the skin hydration level is determined based on the time needed for the current to flow through the skin [83]. The same study enrolled French women over 40 years of age and followed a collagen treatment protocol for three consecutive months [83]. At the end of the study, the authors found a noticeable increase in hydration in the collagen group compared to the placebo group.

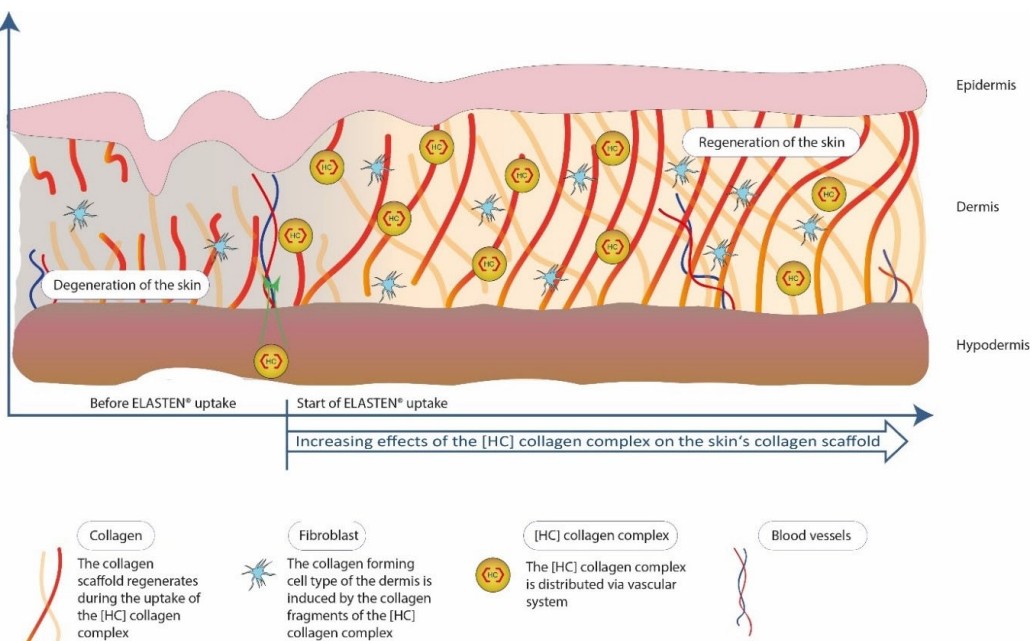

**Figure 12.** Effects of the use of ELASTEN® product (reprinted from [17]).

In another study, women were divided into four groups differing in the dose of collagen they took [81]. Participants were followed up for 60 days, and there was a statistical increase in skin elasticity according to the administered doses compared to the placebo group. Moreover, improvement in skin elasticity among older women was noticed relatively earlier—after just one month [5]. Brittle nail syndrome is a common problem among women, and is characterized by a rough, flaky, jagged surface of the nail. The aim of the study was to check whether daily oral supplementation with collagen peptides alleviates the symptoms of brittle nails and improves their growth rate. The trial involved 25 participants who took 2.5 g of specific collagen peptides (BCP, VERISOL® Gelita, Eberbach, Germany) once a day for 24 weeks. After this period, there was a 4-week break in therapy. The rate of nail growth, as well as the frequency of cracking and overall clinical improvement were assessed by the doctor during and after the therapy. The results of the study showed that treatment with bioactive collagen peptides promoted nail growth by 12% and also reduced the frequency of nail breakage by 42%. Additionally, 64% of study participants reported global clinical improvement in brittle nails and 88% experienced improvement four weeks after treatment. The conclusion of the study is that daily oral consumption of bioactive collagen peptides contributes to improving the rate of nail growth and a noticeable decrease in the frequency of nail breakage [21].

### 4.4. Products with Collagen for Use on the Skin and Mucous Membrane

Collagen found in cosmetics applied to the skin (Figure 13) is not able to replenish its losses in the skin; its role is to retain and absorb water in the outer layer of the skin, i.e., the epidermis [5]. It will also influence the appropriate level of skin hydration. Cosmetic preparations with natural CLG (native collagen, tropocollagen) penetrate the epidermis, where they actively bind water [16,56]; they can also penetrate deeper layers—the dermis—combining there with other water-insoluble proteins, creating a hydrophilic layer on the skin surface, successively retaining moisture in the skin's intercellular spaces [3,71]. CLG is a common ingredient in hydrogel forms, which include widely used beauty masks. Such products are supposed to restore skin elasticity, and have anti-aging properties [39,57]. It can also be used as a thickening agent, but the high price of native CLG limits this choice, and instead gelatin, a denatured form of CLG, is used, which is a much cheaper alternative [15,20]. During the maturation process of collagen, the number of crosslinks in its molecule increases, making it less and less soluble in water and in acidic pH. For

this reason, cosmetic preparations mainly use hydrolyzed peptides, which contain short polypeptides, and small peptides that dissolve well in water. Such hydrophilic molecules are easier to introduce into cosmetic formulations, and thanks to their smaller molecules, they can penetrate deeper layers of the skin and ensure its regeneration [20,33,72]. Based on research on hydrolyzed collagen from marine fish, it was shown that in the deep layers of the skin, the moisturizing properties of hydrolyzed CLG are very high and can have a very good effect on the appearance of the skin [28,56,65,67,74]. Hydrolyzed CLG is also widely known in cosmetics as an ingredient with antioxidant properties and as a natural humectant [39]. Its wide range of application possibilities is due to its good compatibility with human tissues and complete biodegradability. It has been shown that local application and supplementation of hydrolyzed collagen can improve the appearance and properties of the skin by participating in various mechanisms [64] and can also accelerate the wound healing process [40]—in a study in which CLG gel was used in a skin treatment ointment in research group A, higher hydroxyproline content was observed than in the control group that did not use the gel [40,67]. Gel is a form of preparation very often used in dermatological treatment and cosmetics due to its ease of application and non-greasy nature. It is characterized by high water content (95–99%). Its ingredients are water-soluble or form micelles. It also has the ability to create a film, thanks to which it has the ability to maintain active ingredients on the skin surface [69,70,72,84].

Probably, applying collagen gel to the wound stimulates the healthy cells adjacent to it to initiate the healing process [15]. Also important in cosmetic formulas are the film-forming properties of CLG, which are used to create a film on the skin whose task is to reduce TEWL (transepidermal water loss) and protect the skin and hair against damage by surfactants [69,85]. Such properties may be enhanced by binding CLG to biopolymers and/or other polymeric molecules, e.g., polyvinylpyrrolidone (PVP). As a result of the reaction of this compound with CLG, hydrogen bonds are formed. The film-forming properties of collagen can also be modified by substances such as chitosan, silk fibroin, hyaluronic acid, elastin, or keratin [15,20,33,71]. It is important that CLG is introduced into cosmetic preparations after cooling the preparation, at the end of the emulsification process, to avoid the process of protein denaturation. The only completely safe material used in cosmetics, and medicine, is the so-called atelocollagen, i.e., a derivative of type I CLG without telopeptides [67,69]. Research on CLG obtained from the jellyfish "umbrella" has shown that it may have an impact on inhibiting the photoaging process of the skin. Animals were exposed to UVA and UVB rays after being injected with collagen or its hydrolysate in various doses. It was observed that hydrolysate applied topically to the skin protected natural collagen against degradation and also had a protective effect on elastin [61]. Ineffectiveness of topical CLG skin care products may occur due to the inability to penetrate the stratum corneum, which prevents the product from reaching the fibroblasts in the dermis [1,35,40,48]. Therefore, when it comes to collagen products, the chosen route and how they will be used is important [4,34,39,47,56,57,63,73,79]. Much of the research on CLG used for cosmetic applications is carried out by the cosmetics industry, but their results are patented, so there are not many publications in the public domain regarding the uses of collagen.

The other forms of cosmetic products with collagen are creams, serums, or masks. Creams are complex preparations composed of lipophilic and hydrophilic phases. They have the character of an emulsion, i.e., a system consisting of at least two immiscible liquids, one of which is dispersed in the other, creating o/w (oil in water), w/o (water in oil) emulsions or multiple emulsions, e.g., w/o/w. Creams differ in terms of emulsion type, composition, and application [2,5,73]. There can be distinguished creams with a light emulsion formula intended for day use, night creams with more dense and compact consistency, as well as creams in the form of a delicate cream used under the eyes, which are intended for very sensitive skin [71].

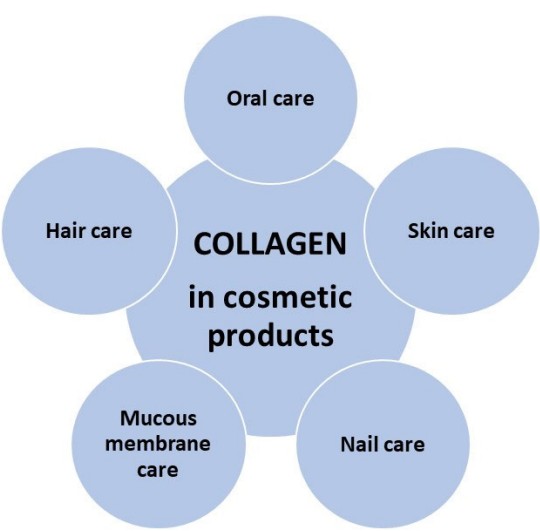

**Figure 13.** Application of collagen in the cosmetic products.

Collagen has been used in creams for decades, so it can be argued that it improves the structure of the skin. However, CLG molecules are too large to penetrate the epidermal barrier when applied to the skin surface. Due to the poor penetration of CLG, cosmetic products use partially hydrolyzed collagen, i.e., amino acids or peptides that would be able to penetrate the skin, but the properties of hydrolyzed CLG differ from those of native one [86]. The larger the peptide (more than 6–7 amino acids), the less likely it is to penetrate the deeper layers of the skin [22]. As a result of CLG hydrolysis, hydrogen bonds are dissociated and the triple helix structure of native CLG changes into a random coil form. Polypeptide chains are also broken, resulting in the formation of a large number of peptides. However, such resulting fragments are unable to reassemble to build CLG in the layers of the skin. Peptides and amino acids have beneficial biological functions such as water retention and cell proliferation, but they are different from those found in CLG itself [86]. In a study in which micronized marine CLG was used in a cream, it was found that it was able to penetrate the stratum corneum. The depth of penetration in human skin was measured using a non-invasive optical device that uses a laser and camera to detect light phase changes that follow the presence of collagen molecules in the deep layers of the skin [86]. In this study, the number of collagen fibers was reduced to facilitate their penetration into the deeper layers of the skin. As a result of the passage through the stratum corneum, the skin's elasticity, texture, and mechanical properties improved. Moreover, after applying the tested collagen cream, the stratum corneum seemed to be better connected to the stratum lucidum/granular layer [86]. Another study focused on the immediate and long-term clinical effects of a topical CLG hydrolysate formulation showed a significant reduction in wrinkles within minutes of the first application, and after three months of use, continued improvement was due to the action of peptides that have been shown to prevent matrix damage, extracellularly induced by metalloproteinases [22].

One other example of a product with collagen is serum. It is a cosmetic preparation in concentrated form, contains higher doses of active substances, and is supposed to act stronger and faster than cream. It can be in the form of an emulsion, gel, milk, or oil and has very light consistency, which facilitates the penetration of active ingredients into the skin. Applying the cream over the serum will prevent it from evaporating. A frequently used ingredient in cheeses is collagen, which improves the elasticity and tension of the skin, as well as hyaluronic acid, plant extracts, and vitamins A, C, and E. In order for the active substances to fulfill their role in the skin, cheeses (and creams) are equipped with carrier systems, i.e., microscopic fat bodies capable of crossing the epidermal barrier [71].

On the cosmetic market there are also masks, which are preparations in the form of a cream or paste, the main role of which is to provide a high dose of nutritional and regenerative ingredients. They are applied to the cleansed skin of the face, neck, or cleavage

and washed off with water or removed completely by removing the elastic layer [34]. There are also masks used for professional treatments in beauty salons, as well as masks for individual use at home. The most frequently used masks are regenerating and nourishing ones, e.g., algae masks, moisturizing or lifting masks aimed at smoothing the skin, reducing wrinkles, and improving skin tension, thereby improving the face oval and minimizing signs of fatigue. Natural collagen hydrolysate is the main ingredient found in the formulas of CLG masks, while CLG flakes soaked in nourishing and moisturizing substances are used to improve the absorption of active substances on facial skin [33,69,71]. CLG masks are also used in anti-aging and moisturizing treatments for people struggling with atopic dermatitis, where they appear as dry flakes made of freeze-dried ingredients applied to the skin, activated by soaking in individual active ingredients. They are supposed to have a lifting effect and retain moisture in the skin [87].

As for the number of studies assessing the effect of local use of collagen products, it is much smaller compared to the studies conducted on oral CLG supplements [5]. A study conducted in 2015 showed [88] that approximately three-quarters of treated women experienced a reduction in wrinkles and a significant improvement in skin density and elasticity after seven days of applying a product containing CLG. A similar study was conducted in Germany and South Africa on 480 patients with wrinkles, sagging skin, scars, and stretch marks, who were administered CLG in transdermal form after preparing the skin with the necessary vitamins and creams for at least a month. The results of the study indicated an improvement in the patients' skin by 60–80% compared to the condition before treatment [88]. Moreover, scientists conducted a histological examination on a subgroup of patients, which showed a significant increase in the amount of CLG and elastin, as well as thickening of the epidermis, mainly the spinous layer, after the first year after the procedure [56,89]. The anti-wrinkle effect of triple peptide (3%) was confirmed by clinicians in the study. The results suggested that their topical application significantly improves the condition of sun-damaged skin (decreases in the number of wrinkles) at the end of the first month of use compared to the placebo group [89]. Previous research shows that both oral and topical use of collagen products reduces or delays skin aging, and there is no evidence that one of these forms is better than the other [5]. One of the critical parameters that accelerate skin aging is the glycation process, especially in cases exposed to environmental factors such as ultraviolet radiation [11,44,50]. Previous research has shown that collagen tripeptide hydrolysate has anti-inflammatory and anti-aging properties, but the exact mechanism of its action is not yet fully known [89]. A 4-week study on the effects of topical application of CLG tripeptide on facial skin was conducted in 22 Asian women with visible wrinkles around the eyes. The age range was from 30 to 54 years. The ampoule used contained hydrolyzed fish skin extract, which is a form of CLG hydrolysate obtained from the skin of Pangasius hypophthalmus, containing 4% Gly-Pro-Hyp with tripeptide content exceeding 25%. The results showed a significant improvement in skin density and elasticity and reduced wrinkles. In addition, a reduction in the accumulation of advanced glycation end products (AGEs) in the skin was demonstrated in the fourth week of the study, without any adverse effects [31]. In vitro studies have shown a preventive effect of local CLG tripeptide on the accumulation of AGEs, on the production of denatured collagen, and on the occurrence of reactive oxygen species in skin fibroblasts. The topical application of collagen tripeptide also resulted in a reduction in the induction of metalloproteinases in the matrix while increasing the level of type I collagen. These study results suggest that the use of topical CLG tripeptide may improve the clinical phenotypes of aging by inhibiting oxidative stress and the glycation process [89]. Other studies have shown that oral and topical use of CLG tripeptide in people with atopic dermatitis results in reduced itching and improved skin hydration due to its anti-inflammatory properties [87,89].

## 5. Side Effects of Overused of Collagen Products

The above examples and discussion showed the positive action of collagen; generally, this peptide is very well tolerated and no adverse effects of oral and topical use have been

observed [5]. However, this does not mean that the use of CLG is absolutely safe. The first effect is allergic reaction, which can be observed in some people. This is the simplest side effect [30]. For example, some people may have a shellfish allergy and could even experience anaphylaxis if they take marine collagen supplements; also, other sources of CLG may cause some allergy problems. In addition, bovine-origin CLG carries a risk of transmitting illnesses such as bovine BSE [30]. Studies have not reported any side effects such as vomiting, diarrhea, nausea, or constipation in treatment or control groups [5]. In his review, Al-Atif reported a wide range of trial studies that found no adverse effects of collagen until they observed their participants [5].

## 6. Conclusions

Collagen has various functions in the body. It maintains the appropriate elasticity of blood vessels, provides mechanical strength to cartilage and bone tissue, is present in the cornea of the eye, and, of course, in the skin, to which it provides appropriate density and elasticity. With age, the production of natural CLG in the skin decreases. Various external factors have a similar effect, such as UV radiation causing photocrosslinking of this protein. The regulation of CLG production involves enzymes called metalloproteinases, whose task is to cut damaged CLG proteins so that new ones can be formed in their place. The most important types of CLG found in the skin are type I and type III. It is an ingredient in various cosmetic products—gels, creams, beauty masks, and serums that accelerate wound healing—and is also a component of oral supplements designed to improve the appearance of skin from its deeper layers. When used in the form of topical products, it is unable to penetrate the epidermal barrier due to its high molecular weight, which means it does not reach fibroblasts in the dermis. For this reason, partially hydrolyzed CLG is used, but its properties will differ from those of native CLG. This protein, after application, remains on the skin surface, preventing water loss, making the skin better hydrated, and, consequently, improving its color, brightness, and overall appearance. It also has a protective role against microorganisms in the case of injured skin. Thanks to this, it has been used in cosmetics for atopic skin. In order for collagen to be included in a cosmetic product, it must be properly extracted and purified. In order to break the strong crosslinks in its molecule, dilute acids and bases are used. Oral supplements used to improve the appearance of the skin and delay its aging process are becoming more and more popular. The site of their action is the dermis, where free amino acids provide building blocks for the synthesis of collagen and elastin fibers, while CLG oligopeptides stimulate fibroblasts to produce new collagen, elastin, and hyaluronic acid. If we want to temporarily improve the appearance of the skin or reduce wrinkles, we should use collagen fillers that have the ability to fill soft tissue defects permanently or temporarily. Numerous studies confirm the effects of CLG preparations as a result of their use: the depth of wrinkles is reduced, the density and elasticity of the skin improves, transepidermal water loss (TEWL) is reduced, and when using oral supplements, it also provides hydration. CLG peptides also prevent damage to the extracellular matrix caused by metalloproteinases. Oral preparations with CLG peptides have an especially positive effect on nails; a significant reduction in nail brittleness during and after using a preparation was observed. The presented work certainly does not exhaust such an extensive topic as the use of collagen protein in cosmetics. It is only an attempt to demonstrate what a huge role this protein plays in the processes taking place in the body, and that the largest organ of the body, which is the skin, would not be able to function properly without its presence. The cosmetic market offers a wide range of collagen products, which was confirmed in this work, but it is important how we use them (topically, orally, or by injection), and this depends on the needs of the skin.

To end, it is worth mentioning potential avenues for future research. These could include exploring novel formulations, investigating the long-term effects of collagen use, or delving into the molecular mechanisms underlying collagen's impact on skin health.

**Author Contributions:** Conceptualization, B.J.; methodology, B.J. and T.O.; formal analysis, B.J. and T.O.; investigation, B.J. and Z.M.; data curation, B.J. and Z.M.; writing—original draft preparation, B.J., Z.M. and T.O.; writing—review and editing, B.J. and T.O.; visualization, B.J.; supervision, B.J. All authors have read and agreed to the published version of the manuscript.

**Funding:** This research received no external funding.

**Institutional Review Board Statement:** Not applicable.

**Informed Consent Statement:** Not applicable.

**Data Availability Statement:** Publicly available publications were analyzed in this study. All used sources are included in the list of references.

**Conflicts of Interest:** The authors declare no conflicts of interest.

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
