# Peer review of "Use of Collagen in Cosmetic Products"

_cimb, doi:10.3390/cimb46030132_

Round 1

Reviewer 1 Report

Comments and Suggestions for Authors

The paper on the 'Use of Collagen in Cosmetic Products' offers a thorough exploration of collagen's structural attributes, its natural presence in the dermis, and its application in various cosmetic formulations. While providing valuable insights into the effects of collagen-containing products on skin health and aging, the paper could benefit from a more in-depth analysis of collagen types, quantitative data on efficacy, and consideration of potential side effects for a more nuanced and comprehensive discussion. The review is written well and related studies are presented comprehensively, however, there are major areas which needs to be improved before decision is made. My comments are given below-

  • In-Depth Analysis of Collagen Types: While the paper briefly mentions different types of collagen, a more in-depth analysis of each type and its specific properties would add value. This could include a discussion on collagen I, II, and III, elucidating their unique roles in skin health.
  • Mechanisms of Action: The paper could benefit from a more detailed discussion on the mechanisms by which collagen, or its derivatives, interact with the skin. Elaborating on how collagen influences cellular processes and signaling pathways would provide a more mechanistic understanding of its cosmetic effects.
  • Quantitative Data on Efficacy: Including quantitative data from studies assessing the efficacy of collagen-containing products would strengthen the paper. For instance, presenting statistical analyses, such as p-values and effect sizes, would provide a more rigorous foundation for the reported improvements in skin condition.
  • Consideration of Age and Skin Types: The paper should acknowledge potential variations in the effectiveness of collagen products based on age and different skin types. This consideration would offer a more nuanced perspective on the generalizability of the reported benefits.
  • Discussion on Potential Side Effects: An inclusion of potential side effects or limitations associated with the use of collagen-based cosmetic products would make the paper more balanced. Addressing any known drawbacks or considerations for specific populations would enhance its practical relevance.
  • Future Research Directions: To foster further scientific inquiry, the paper could conclude with a section suggesting potential avenues for future research. This could include exploring novel formulations, investigating the long-term effects of collagen use, or delving into the molecular mechanisms underlying collagen's impact on skin health.
  • Comparison with Alternative Ingredients: A comparative analysis with alternative ingredients commonly used in cosmetic products would add depth to the discussion. Exploring how collagen measures up against other substances with similar purported benefits could help guide both consumers and researchers.
  • Citation of Recent Studies: To ensure the paper reflects the latest scientific advancements, consider including more recent studies and findings. This would demonstrate the authors' awareness of the evolving landscape in collagen research and cosmetic science. For example cite https://doi.org/10.1007/s44174-023-00087-8 with the sentence ‘Its presence, e.g. in bone tissue, provides elasticity and strength.’ Cite https://doi.org/10.1021/acs.langmuir.2c00671 with the sentence ‘…..graded and the more active groups are exposed to interactions with water through hydrogen bonding’ to make references up to date.
  • Clarification on Unusual Roles of Collagen: The paper mentions collagen's unusual roles in the human body without specifying these roles. Providing more detail on these lesser-known functions would contribute to a more comprehensive understanding of collagen's significance.
  • Language Clarity: Ensure that the language used is clear and accessible to a broad audience. Avoid unnecessary technical jargon and, when introduced, provide explanations to enhance the paper's accessibility to a wider readership
Comments on the Quality of English Language

minor

Author Response

I attached the file with responses

Reviewer 2 Report

Comments and Suggestions for Authors

The manuscript entitled "Use of collagen in cosmetic products" is a medium-size review starting with information regarding the various collagen types and their features, followed by presentation of the collagen role in the skin, and continuing with the various collagen products used in cosmetics. It is a useful source of information and can contribute to the field of the cosmetics' research and applications. However, some revisions are necessary, so as to improve the quality of the manuscript:

1) Subsections "2.6. Animal sources" and "2.7. Sources of collagen used in cosmetics" do not belong to the section "2. Characteristics of collagen". I propose to be transferred as separate subsections to "4. Characteristics of cosmetic products containing collagen"

2) There is a certain redundancy in the text. Some information is repeated many times throughout the manuscript, e.g. in which tissues collagen can be found (lines 44, 78, Table 1) or that collagen type I can be found in bones (lines 124 and 126) or that heterogeneity of collagens is the result of post-translational modifications (lines 115 and 128). The authors should modify their text, so as to avoid repetitions.

3) In lines 103-104, the sentence should be amended, in order to be clear that X and Y may represent any amino acid residue, although the frequency of Pro and Pro-OH is high (as indicated in the text).

Comments on the Quality of English Language

There are several language and/or grammar errors. I am quoting some of them indicatively:

Lines 39-40 should read "themselves" instead of "the selves"

Lines 40-41 should read "depending" instead of "d pending"

Line 42 should read "structure" instead of "struture"

Line 44 should read "It occurs" instead of "Occurs"

In line 919, Ref. 4, the word "by" should be eliminated

Author Response

I attached the file with responses

Round 2

Reviewer 1 Report

Comments and Suggestions for Authors

accept

Reviewer 2 Report

Comments and Suggestions for Authors

The revised manuscript is suitable for publication.

Comments on the Quality of English Language

There are no comments on the manuscript's language quality.